# Quercetin in Animal Models of Alzheimer’s Disease: A Systematic Review of Preclinical Studies

**DOI:** 10.3390/ijms21020493

**Published:** 2020-01-13

**Authors:** Xiao-Wen Zhang, Jia-Yue Chen, Defang Ouyang, Jia-Hong Lu

**Affiliations:** State Key Laboratory of Quality Research in Chinese Medicine, Institute of Chinese Medical Sciences, University of Macau, Taipa, Macao; xiaowen_zhang10@163.com (X.-W.Z.); chenjy576@gmail.com (J.-Y.C.); defangouyang@um.edu.mo (D.O.)

**Keywords:** Alzheimer’s disease, quercetin, neuroprotective effects, animal AD models

## Abstract

Alzheimer’s disease (AD) is the leading cause of dementia worldwide. It involves progressive impairment of cognitive function. A growing number of neuroprotective compounds have been identified with potential anti-AD properties through in vitro and in vivo models of AD. Quercetin, a natural flavonoid contained in a wide range of plant species, is repeatedly reported to exert neuroprotective effects in experimental animal AD models. However, a systematic analysis of methodological rigor and the comparison between different studies is still lacking. A systematic review uses a methodical approach to minimize the bias in each independent study, providing a less biased, comprehensive understanding of research findings and an objective judgement of the strength of evidence and the reliability of conclusions. In this review, we identified 14 studies describing the therapeutic efficacy of quercetin on animal AD models by electronic and manual retrieval. Some of the results of the studies included were meta-analyzed by forest plot, and the methodological quality of each preclinical trial was assessed with SYRCLE’s risk of bias tool. Our results demonstrated the consistent neuroprotective effects of quercetin on different AD models, and the pharmacological mechanisms of quercetin on AD models are summarized. This information eliminated the bias of each individual study, providing guidance for future tests and supporting evidence for further implementation of quercetin into clinical trials. However, the limitations of some studies, such as the absence of sample size calculations and low method quality, should also be noted.

## 1. Introduction 

Alzheimer’s disease is the leading cause of dementia worldwide, involving progressive impairment of cognitive function that severely affects daily living. AD is characterized classically by two hallmark pathological features: β-amyloid (Aβ) plaque deposition and neurofibrillary tangles (NFT) accumulation. To date, available symptomatic treatments, such as cholinesterase inhibitors and memantine, have helped improve quality of life, but they have failed to reverse the course of disease or slow down the rate of cognitive function decline [1]. There is a huge need for novel drug investigations in both preclinical and clinical studies.

Quercetin (3,3′,4′,5,7-pentahydroxyflavone) is a kind of bi-flavonoid distributed in multiple fruit and vegetable species; thus, it is constantly consumed by people in daily life. Quercetin has multiple pharmacological applications, including antioxidant, neuroprotective, anti-viral, anti-cancer, anti-microbial, anti-inflammatory, hepatoprotective, and anti-obesity activities [2]. Various studies have been carried out to investigate the neuroprotective effects of quercetin in the central nervous system, especially in multiple in vitro and in vivo models of AD. For example, Nakagawa et al. reviewed seven studies of animal experiments in the past few years evaluating the neuroprotective effect of quercetin, in which quercetin dramatically improved cognition and memory deficits in rodent animal AD models [3].

However, each independent study may have had limitations in methodology or the theoretical basis of its study design, and these individual animal experiments provided different viewpoints regarding the pharmacological mechanisms of the promising drug for AD [4]. This systematic review used a scientific approach to eliminate the bias of independent publications and provides a comprehensive introduction to the efficacy of quercetin on experimental animal AD models with different pathological features. The systematic summarization of the drug’s pharmacological mechanism also supplied a clear map to focus on in future drug studies. Furthermore, the evaluation based on animal experiments data gave an objective judgement of the strength of evidence and the reliability of conclusions, which may guide the design of future preclinical tests and give supporting evidence for quercetin as a promising drug in clinical treatment.

## 2. Methods 

### 2.1. Search Strategy

We performed comprehensive literature retrieval to identify articles investigating the neuroprotective effects of quercetin in rodent animal AD models. We collected the relevant publications from three independent databases: PubMed, Web of Science, and Google Scholar. We used “Alzheimer’s disease” AND “Quercetin” as the key words for the literature search. The search was not restricted by date or language. In Google Scholar, only the articles whose titles included the key words were selected, in order to achieve a fairly accurate retrieval. The reviewer (Xiao-Wen Zhang) evaluated the qualifications of animal studies independently according to the eligibility criteria by screening the abstracts of the identified publications.

### 2.2. Inclusion Criteria

#### 2.2.1. Inclusion Criteria:

1. Laboratory rodents of any species, age, sex, or weight producing AD models were covered.

2. Any kind of comparison between quercetin intervention and the control group was included. The control group had to comprise a placebo, such as physiological saline or a similar vehicle. Dosage, administration route, and therapy duration of the drug were not limited.

3. Original experimental studies to measure the efficacy of quercetin on AD animal models were included.

#### 2.2.2. Exclusion Criteria:

1. Duplicated references; review articles; lack of full text; literature with incorrect or incomplete data.

2. Not evaluating the effects of quercetin on rodent animal AD models.

3. The quercetin was tested on rodent animal AD models in combination with other chemicals or interventions. 

### 2.3. Data Extraction and Quality Assessment 

The detailed information of the articles included is listed in the following way (Table 1): (1) author and publication year; (2) animal data, including modeling methods, species, gender, age, and weight; (3) drug usage, including dosage, administration route, and therapy duration; (4) outcome measure; (5) neuroprotective activities and mechanisms. Part of the results of studies were meta-analyzed by forest plot with RevMan 5.3 (Cochrane Community, London, UK).

We used the SYRCLE’s Risk of Bias (RoB) tool that is specifically designed for animal experiments to assess the methodological quality of the publications. The criteria included: random allocation sequence; similar baseline characteristics; allocation concealment; random housing; blinded intervention; random selection for outcome assessment; blinded assessment of outcome; incomplete outcome data; selective outcome reporting; other sources of bias. A quality score out of a possible total 10 points was given to each included study.

## 3. Results 

### 3.1. Study Selection 

Initially, a total of 252 references were identified, 204 of which were found via Pubmed and Web of science databases, and 48 were from Google Scholar. By browsing titles and abstracts, we then selected 24 full-text articles about animal studies. We read the whole texts of the 24 references to assess their eligibility. Finally, 14 qualified articles were included in the systemic review for meta-analysis (Figure 1). It was notable that eight of the total 14 articles (57%) were published in the past three years, indicating an increasing interest on the anti-AD properties of quercetin.

### 3.2. Study Characteristics

#### 3.2.1. Different Animal Models

The 14 studies involved five main kinds of AD models. They were Aβ-injection, lipopolysaccharide (LPS)-induced, pentylenetetrazole (PTZ)-induced, senescence accelerated mouse (SAM), and APP/PS1/tau-transgenic AD models. The Aβ injection model can induce Aβ accumulation and deposition in the brain, but as an acute toxicity model it cannot reproduce the progressive neurodegeneration process [18]. The drug-induced models (LPS and PTZ) display neurodegeneration and cognitive impairments, but they lack AD-related histological hallmarks [13,14]. Senescence-accelerated mice belong to a naturally rapidly aging species, and this accelerated aging-induced dementia model can recapture the cognitive impairment and pathological features that are closer to actual AD patients [11].

In addition to toxin-induced models, transgenic mice are also powerful tools in a disease study. Based on the discovery of familial Alzheimer’s Disease (FAD) mutated genes, transgenic AD models consisted of APP23 mice, APP_swe_/PS1_dE9_ mice, 3×Tg-AD mice (PS1_M146V_ knock-in, APP_swe_, tau_p301L_), and 5×FAD mice (APP695 harboring Swedish, Florida, London mutations, and PS1 harboring M146L, L286V mutations) have been extensively developed. The APP/PS1/tau-transgenic mice models replicate the Aβ accumulation and NFT formation in the brain, and display neurodegeneration and cognitive dysfunction. They are ideal for studying the roles of APP and tau in the occurrence and development of AD. Considering that more than five included articles proved the effect of quercetin on attenuating Aβ aggregation, the transgenic mice models have become the mainstream for preclinical pharmacodynamic studies on AD. To better understand the animal AD models with different pathological features and their application in the therapeutic trails, we summarized these related models as follows (Table 2).

#### 3.2.2. Behavioral Test Analysis

The Morris water maze (MWM) test is a critical parameter task to evaluate spatial learning and memory functions, playing an important role in assessing the levels of cognitive function and memories of AD models and evaluating the drug’s efficacy. The probe test of MWM is performed after days of training by removing the platform and allowing testing animals to swim freely for 60 s. The number of times one crosses the target quadrant (where the platform is located during training days) and the percentage of time it spends in the quadrant are considered to represent the degree of memory consolidation having taken place after training [13].

There were nine studies conducting the MWM test, only four of whom recorded the number of times crossing the target quadrant, while the other five did not. Here, we used ImageJ (Rawak Software Inc., Stuttgart, Germany) and RevMan 5.3 (Cochrane Community, London, UK) to quantify and analyze the data of four studies. We meta-analyzed the outcome data by forest plot to assess the protective effect of quercetin. A total of 45 mice AD models were treated with quercetin in different dose and time manners (20–30 mg/kg, from 2 weeks to 3 months), and another 53 mice in the control group were correspondingly treated with vehicle.

The results are shown in the following forest plot (Figure 2). In the experimental group administered with quercetin, the number of times crossing the target quadrant was distinctly increased compared to this in the vehicle treated group (*p* < 0.00001), with a mean differences of 4.39 in the subgroup of transgenic AD mice, 4.57 in the subgroup of LPS-induced mice, and 4.44 in the total outcome. These data suggested a constant effect of quercetin on ameliorating learning and memory functions throughout the different studies. Furthermore, there was no heterogeneity in the subgroup of transgenic AD mice (*p* = 0.58, I^2^ = 0%). The analysis of the combination outcome also indicated that quercetin treatment remarkably improved the cognitive function of AD mice (*p* < 0.00001), with no significant heterogeneity (*p* = 0.77, I^2^ = 0%).

#### 3.2.3. Neuroprotective Mechanisms Analysis

This systematic review gives a general introduction to the neuroprotective mechanisms of quercetin in all the included studies, mainly involving the inhibition of Aβ aggregation and tau hyper-phosphorylation, anti-oxidative activity, and anti-inflammatory activity. Besides, there were also articles which revealed the effects of quercetin on ameliorating mitochondrial dysfunction.

Six studies indicated that quercetin displayed the inhibitory effect on Aβ plaque aggregation and neurofibrillary tangles formation, the most important histological hallmarks of AD. Wang DM. et al. [6] reported that quercetin-treated AD mice exhibited far fewer amyloid plaques and a lesser plaque area in the hippocampus and cortex. Sabogal-Guáqueta AM. et al. [8] reported a remarkable reduction in the extracellular Aβ plaque amount, the Aβ1-40 and Aβ1-42 levels, and the C-terminal APP fragments (β) in the hippocampi of mice treated with quercetin. This study also demonstrated that quercetin administration could reduce the paired helical filament (PHF) and AT-8 (anti-tau pSer202/Thr205) protein levels in hippocampal and amygdala lysates; and decrease the expression of hyperphosphorylated tau in the subiculum, CA1 area, and amygdala. Zhang X. et al. [9] reported that the insoluble Aβ40 and Aβ42 levels, and the plaque burden were significantly reduced in the cortexes of quercetin-treated 5×FAD mice. Their data suggested that quercetin increase the levels of apolipoprotein E (apoE), which is pivotal for cerebral cholesterol metabolism and clearance of Aβ, via inhibiting apoE degradation in astrocytes. Lu Y. et al. [15] indicated that in the early-middle stage of AD mice with quercetin enrich diet, the protein level of β-site APP cleaving enzyme 1 (BACE1) was decreased to depress Aβ plaque production, but a similar effect of quercetin was not observed in the middle-late stage of AD mice. Paula PC. et al. [17] reported that quercetin treatment strongly prevented Aβ aggregation and partially decreased hyperphosphorylated tau in the CA1 region of the hippocampus and in the amygdala, without changes in the entorhinal cortex of AD mice. Li Y. et al. [18] showed that quercetin treatment to AD rats reduced the β-amyloid accumulation in the CA1 region of hippocampus.

Three of the total 14 studies assessed Aβ42 levels in mice’s brains by ELISA. Fifteen transgenic AD mice received different doses and duration of quercetin (20 mg/day for 4 weeks, 25 mg/kg/2 days for 3 months, and 500 mg/kg/day for 10 days) while 17 were correspondingly treated with vehicle. A standardized mean difference (SMD) was used to eliminate the deviation because each independent study measured the same outcome index with different units. As shown in the following forest plot (Figure 3), the quercetin treated group presented a remarkable decline in Aβ42 levels measured by ELISA compared with the control group (*p* = 0.0005), and there was a very moderate heterogeneity in the outcome (*p* = 0.03, I^2^ = 73%).

Meanwhile, three included studies measured Aβ levels in the CA1 regions and cortexes of mice’s brains by immunohistochemistry (IHC). The following meta-analysis (Figure 4) indicated that compared with the vehicle control group, Aβ immunoreactivity was significantly decreased in the quercetin treated group (*p* < 0.00001), with mean differences of −3.51 and −8.55 in the subgroups, and −4.68 in the overall outcome. In addition, there was a mild heterogeneity in the subgroup of Aβ immunoreactivity in CA1 region (I^2^ = 82%, *p* = 0.004) and in the total combination (I^2^ = 77%, *p* = 0.002).

Three studies examined the anti-oxidative function of quercetin on animal AD models. Superoxide dismutase (SOD) and glutathione (GSH) are important antioxidant enzymes that prevent the brain tissue from oxidative injury. Malondialdehyde (MDA) is produced from lipid peroxidation and generally regarded as a marker of oxidant injury. Rishitha N. et al. [14] reported that the solid lipid nanoparticle of quercetin (SLN-Q) had an ameliorative effect on the decreased GSH levels in AD zebrafish brains in a dose-dependent manner. Li Y. et al. [18] indicated that in quercetin-treated mouse brains, MDA was decreased, whereas SOD and GSH were increased compared with the vehicle control group. Their data suggested that quercetin exhibited protective effects against oxidative stress mediated neuronal damage by modulating the expression of Nrf-2 dependent antioxidant responsive elements. Nrf-2 is a transcription factor balancing the gene expressions of many familiar antioxidant enzymes to modulate the antioxidant response, and quercetin induced Nrf-2 translocation from cytoplasm to nucleus to activate the expression of various protective genes. Patil CS. et al. [5] reported that quercetin prevented the oxidant protease-mediated tissue injury in AD mice via inhibiting cyclooxygenase-2, proteases, free radicals, and nitric oxide production.

Five studies indicated that quercetin had the anti-inflammatory effect by suppressing activated astrocytosis and microgliosis. In the central nervous system, microglia and astrocytes are important modulators of neuroinflammation, responding rapidly to complaints such as infection, stress, and injury. Studies have proven that in the pathological development of AD, activated microglia and astrocytes are responsible for the neuroinflammation-mediated neurodegeneration. Glial fibrillary acidic protein (GFAP) is a major intermediate filament protein specific to astrocytes and Iba-1 is a specific marker for activated microglia. Sabogal-Guáqueta AM. et al. [8] and Khan A. et al. [13] reported that quercetin significantly reduced GFAP and Iba-1 immunoreactivity in the CA1 hippocampal areas, the cortexes, and the amygdalas of AD mice compared to vehicle treatment. Moreno LCGEI. et al. [11] demonstrated that GFAP expression was strongly decreased in the hippocampi of SAMP8 mice treated with quercetin-loaded nanoparticles. Lu Y. et al. [15] reported that AD mice with quercetin enriched diets exhibited a decrease in protein levels of *p*-Smad2 and *p*-STAT3, which play critical roles in activation of astrogliosis. Besides, Khan A. et al. [13] also indicated that quercetin halted the activated TLR4/NFKB pathway which is responsible for inflammatory signaling, and tempered the expressions of several inflammatory mediators, such as TNF-α, COX-2, NOS-2, and IL-1β in AD mice’s cortexes and hippocampi. Vargas-Restrepo F. et al. [12] reported that quercetin relieved both astrocytic and microglial activation induced by modeling, and quercetin-treated mice presented a significant reduction in iNOS and COX-2 immunostaining in the CA1 area.

In addition, two articles presented the protective effects of quercetin in ameliorating mitochondrial dysfunction. Mitochondrial dysfunction could result in decreased mitochondrial membrane potential (MMP), impaired ATP syntheses, and enhanced ROS production, which usually lead to mitochondrial damage accumulating with age. Wang DM. et al. [6] reported that treatment with quercetin significantly attenuated mitochondrial damage through enhancing MMP, ATP, and reducing ROS expression levels in the hippocampal mitochondria of AD mice. In addition, Khan A. et al. [13] indicated that quercetin prevented the neuronal degeneration mediated by mitochondrial apoptotic pathway. Their data suggested that the anti-apoptotic Bcl-2 and pro-apoptotic Bax markers play as an important indicator in the mitochondrial apoptotic pathway, and the increased Bcl-2/Bax ratio can induce the over-activation of Cytochrome C, a primary mediator in the mitochondrial associated pathway. Quercetin exhibited protective effects against the apoptotic neurodegeneration, significantly reducing cytochrome C’s expression level and Bax/Bl2 ratio. The neuroprotective mechanisms of quercetin in treating AD are summarized in Figure 5.

### 3.3. Methodological Quality Assessment

Animal experiments play an important role in bridging the gap between basic research and clinical trials, and the risk of bias is a critical parameter for evaluating the methodological quality in study design, drug administration, and results extraction. Therefore, we used the SYRCLE’s risk of bias tool to evaluate the methodological quality of 14 studies according to the instructions of Hooijmans CR. et al. [19]. “Yes” indicates a low degree of bias in this quality item, and “no” indicates a high risk of bias. “Unclear” indicates data supplied was insufficient to evaluate the bias degree.

The quality items score of each study was ranged from two to six out of a total 10 points. Random allocation was only reported by three publications (21.4%); random selection for outcome assessment by only four publications (28.6%); blind intervention and blind outcome assessment were not clearly described. There were mainly three types of bias among the 14 included studies: bias due to inadequate randomization and lack of blinding; reporting bias which is difficult to assess because protocols of most animal intervention studies are not publicly accessible; other bias such as experimental unit-of-analysis errors. The high risk of bias could harm the internal validity of evidence from AD animal studies, which would subsequently slow the efficiency of translating preclinical trial outcomes into clinical practice, regarding quercetin treatment. In addition, sample size calculation was not mentioned in any publication. The sample size of an animal study is supposed to be large enough to find out the biologically significant difference, but meanwhile, small enough to avoid the unnecessary sacrifice. The details of quality assessment are displayed in Table 3.

## 4. Discussion

In recent years, active, small molecule compounds from natural plants have been robustly studied for their pharmacological efficacy in treating the neurodegenerative diseases. For example, Habtemariam S. et al. [20] reviewed rutin (quercetin-3-O-rutinoside), a multifunctional natural flavonoid glycoside, as a natural therapy for AD, and its mechanisms; Zhu Q. et al. [21] reviewed the neuroprotective effects of baicalein, a bioactive flavone component from traditional Chinese medicine, in animal models of Parkinson’s disease. Quercetin, a ubiquitous flavonoid widely contained in vegetables and fruits, is generally classified as an enhancer of cognitive performance in traditional and oriental medicine. Different in vitro and in vivo studies have displayed the therapeutic efficacy of quercetin against neurodegenerative disorders. The meta-analysis in the systemic review used a methodical approach to eliminate the bias of each individual study, very helpful for evaluating the drug’s effect in individual preclinical trials. Here, we systematically analyzed the protective effects of quercetin in different animal AD models, which provided unbiased evidence for further implementation of quercetin as a candidate drug in AD clinical treatment.

A reliable AD experimental model is very important for the exploration of the pathogenesis and evaluation of the therapeutic efficacy of the intervention. Of the total 14 publications included, most toxin-induced AD models did not recapture the progressive neurodegeneration process under acute toxicity; some lacked AD-related Aβ deposition and NFT formation. The transgenic mice models harboring the concrete transgenes to simulate AD pathophysiology seem to be more proper tools for drug research. Esquerda-Canals G. et al. [22], in a review, concluded that these transgenic mouse lines usually harbored mutated human genes at several loci, such as APP, PSEN1, and APOE 4. Therefore, not only the aggregation of Aβplaque but also cholesterol and insulin metabolism are emulated.

Three of the total 14 studies used nanoparticles of quercetin rather than free solution as the drug’s delivery mode. Sun D. et al. [10] designed PLGA-functionalized quercetin nanoparticles to facilitate the controlled delivery of quercetin into brain. Another two articles also evaluated the therapeutic efficacy of solid lipid nanoparticles of quercetin and quercetin-loaded in zein nanoparticles on experimental AD models [11,14]. In fact, the clinical approach of quercetin utilization is highly hampered by its low oral bioavailability (about 2%), which is due to the low aqueous solubility and partial permeability of blood-brain barrier (BBB). Recently, nanoparticle formulation has played a promising role in crossing the BBB and producing target specific action. Encapsulation of quercetin in nanoparticles (as oral carriers) is suggested to distinctly improve its delivery efficacy into brain, thereby better ameliorating the cognitive disorders in preclinical and clinical trials [23].

In this review, the outcome data of Morris water maze test from four publications were meta-analyzed by forest blot to evaluate the effects of quercetin on attenuating spatial learning and memory dysfunction. The statistical results presented a significant improvement in the cognitive performance of quercetin-treated AD models, and there was no distinct heterogeneity among the independent studies. In addition, the protective mechanisms of quercetin were systematically introduced, mainly involving the inhibition on Aβ aggregation and tauopathy; the anti-oxidative and anti-inflammatory activity; and amelioration of mitochondrial dysfunction. Zaplatic E. et al. [24] also reviewed the molecular mechanisms underlying protective role of quercetin in attenuating AD in both in vivo and in vitro studies. In addition to above-mentioned aspects, this article reported the antioxidant potential of quercetin in modulating ERK1/2, PI3K/Akt, JNK, and MAPK signaling pathways and gene expressions.

Overall, we comprehensively reviewed 14 animal studies assessing the protective effect of quercetin in AD models, which were strictly selected from three separate databases based on above-mentioned inclusion criteria. The outcome data from included studies supplied convincing evidence for the anti-AD efficacy of quercetin in preclinical trials. We also systematically introduced the pharmacological mechanisms and assessed the methodological quality of each study, which could be very helpful for instructing future tests or avoiding unnecessary replications. However, the limitations of this review should also be noticed: the negative outcome of a study may not be completely reported. The methodical quality of some publications is very low. Required items, including randomized allocation, randomized selection for assessment, and blinding methods, may not have been strictly conducted; and sample size calculations of all publications are not mentioned. This might lead to biased information in the conclusion.

## Figures and Tables

**Figure 1 ijms-21-00493-f001:**
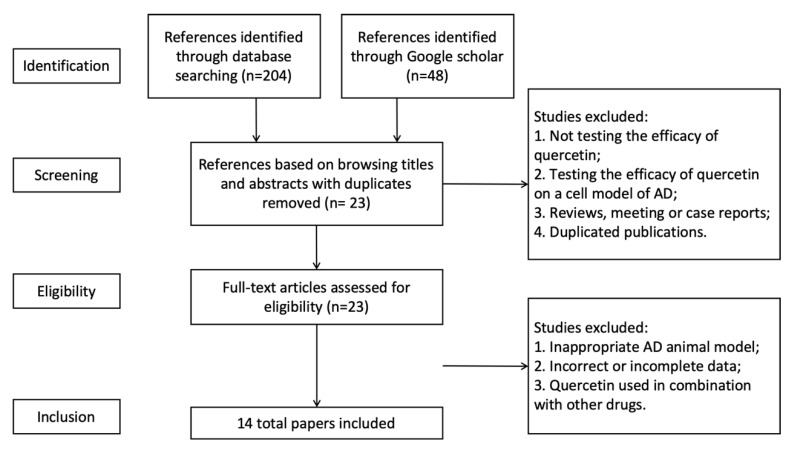
Selection methodology for study inclusion.

**Figure 2 ijms-21-00493-f002:**
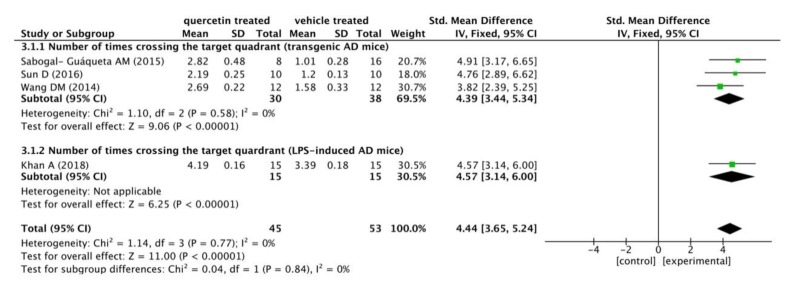
Forest plot for MWM test analysis: quercetin versus vehicle control. The mean difference and standard error of the number of times crossing the target quadrant in the MWM test of each study was quantificationally measured by ImageJ. The data were meta-analyzed in the form of a forest plot by RevMan 5.3. Based on their different animal models, the four studies were divided into two subgroups.

**Figure 3 ijms-21-00493-f003:**
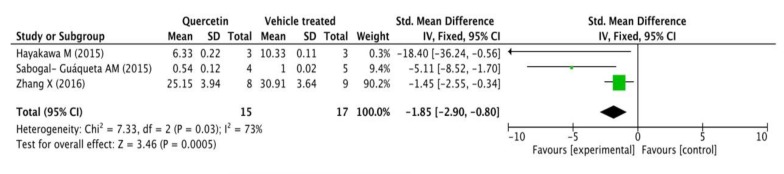
Forest plot for Aβ42 measured by ELISA analysis: quercetin versus vehicle control. The mean difference and standard error of Aβ42 level (ELISA) in the quercetin and vehicle treated groups were quantificationally measured by ImageJ. This data was meta-analyzed in the form of forest plot by RevMan 5.3.

**Figure 4 ijms-21-00493-f004:**
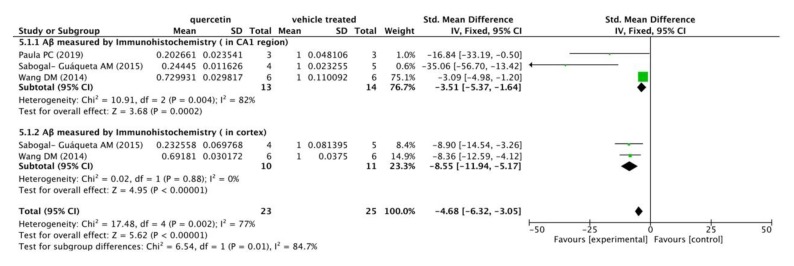
Forest plot for Aβ measured by immunohistochemistry analysis: quercetin versus vehicle control. The mean difference and standard error of Aβ level (IHC) in the quercetin and vehicle treated groups of each study were quantificationally measured by ImageJ. This data was meta-analyzed in the form of forest plot by RevMan 5.3. The studies were divided into two subgroups based on Aβ distribution in mice’s brains (CA1 region and cortex).

**Figure 5 ijms-21-00493-f005:**
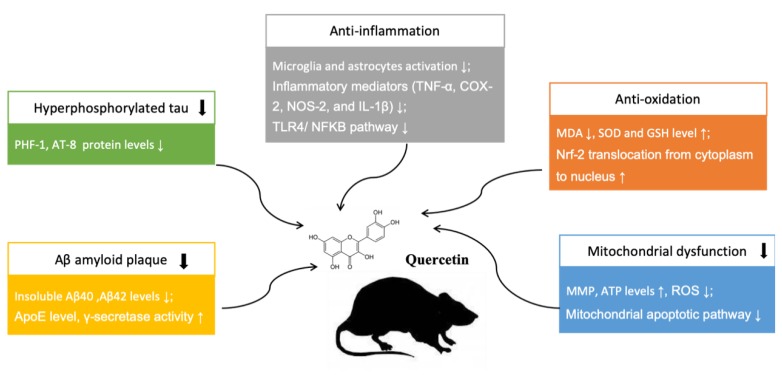
Neuroprotective mechanisms of quercetin in animal AD model. “↑” means up-regulation, “↓” means down-regulation.

**Table 1 ijms-21-00493-t001:** Detailed information of the 14 studies.

Author (Year)	Animal Data	Quercetin Administration	Outcome Measure	Pharmacological Activities (Mechanisms)
Patil CS (2003) [5]	LPS-induced mice AD model (Swiss mice, male&female, 3 months old, 15–20 g and 16 months old, 35–40 g)	Dosage: 25, 50, and 100 mg/kg/day; Ad: i.p.; Duration: 7 days	Behavioral test (elevated plus maze, locomotor activity test, passive avoidance task, Rota-Rod test)	Prevented the cognitive impairment (oxidative stress↓)
Wang DM (2014) [6]	APP_swe_/PS1 _dE9_ transgenic AD mice (C57/BL) (male&female, 3 months old)	Dosage: 20 and 40 mg/kg/day; Ad: p.o.; Duration:16 weeks	Behavioral test (Novel Object Recognition Test, Morris Water Maze); Thioflavine S staining (Aβ deposition); WB (AMPK and p-AMPK levels)	Lessened cognitive deficits, reduced Aβ plaques and ameliorated mitochondrial dysfunction (AMPK activity↑)
Hayakawa M (2015) [7]	APP23 AD mice model (8 weeks old)	Dosage: 20 mg/day; Ad: p.o.; Duration: 4 weeks	Behavioral test (Contextual and auditory fear conditioning test); WB and ELISA (Aβ1-42, GADD34, ATF4, eIF2 a, etc.)	Improved memory (p-eIF2 a↓ and ATF4↓ through GADD34 induction)
Sabogal-Guáqueta AM (2015) [8]	Homozygous 3 xTg-AD mice (male&female, 18-21 months old)	Dosage: 25 mg/kg/2 days; Ad: i.p.; Duration: 3 months	Behavioral test (Elevated plus maze, Morris Water Maze); Immunohistochemistry (NeuN, βA, AT8, GFAP and Iba-1); WB (AT8, tau 5 and PHF-1 levels); ELISA (CTFα, CTFβ and βA1-40,42)	Reversed histological hallmarks of AD and protected cognitive and emotional function
Zhang X (2016) [9]	5XFAD transgenic mice (male&female, 6–8 weeks old)	Dosage: 500 mg/kg/day; Ad: p.o.; Duration: 10 days	Immunohistochemistry for Aβ; WB and qRT-PCR for apoE; ELISA (Aβ40 and Aβ42)	Increased brain apoE and reduced insoluble Aβ levels (inhibited apoE degradation)
Sun D (2016) [10]	APP/PS1 transgenic AD mice	Dosage: 10, 20 and 30 mg/kg (PLGA@QT NPs); Ad: i.v.; Duration: 30 days	Behavioral test (Morris Water Maze, Novel Object Recognition Test)	PLGA-functionalized quercetin (PLGA@QT) NPs ameliorated cognition and memory impairments
Moreno LCGEI (2017) [11]	SAMP1&SAMP8 mice (Male, 5 months old, 28–30 g)	Dosage: 25 mg/kg/day (Q) and 25 mg/kg/2 days (NPQ); Ad: p.o.; Duration: 2 months	Behavioral test (Morris Water Maze, Open field test, Rotarod test, Marble burying test); WB (GFAP, CD11)	Nanoencapsulaed quercetin (NPQ) improved the cognition and memory impairments (GFAP↓)
Vargas-Restrepo F (2018) [12]	Homozygous 3xTg-AD mice (male&female, 18–21 months old)	Dosage: 25 mg/Kg/48 h; Ad: i.p.; Duration: 3 months	Immunofluorescence (Iba-1 and βA); immunohistochemistry (GFAP, iNOS and COX-2)	Anti-inflammatory effect in CA1 hippocampal region
Khan A (2018) [13]	LPS-induced mice AD model (male, 8 weeks old, 25–30 g)	Dosage: 30 mg/kg/day; Ad: i.p.; Duration: 2 weeks	Behavioral test (Morris Water Maze, Y-maze); WB (GFAP, Iba-1, TLR4/NFKB, TNF-α, Caspase-3, etc.); Immunofluorescence (GFAP, Iba-1, p-NFKB, IL-1β, Caspase-3, etc.); Nissl staining	Reduced gliosis, prevented neuroinflammation in cortex and hippocampus, rescued the mitochondrial apoptotic pathway and neuronal degeneration (cyto. C↓, caspase-3↓ and PARP-1↓)
Rishitha N (2018) [14]	PTZ-induced Zebrafish AD model (adult male, <8 months old, 1.0–1.2 g)	Dosage: 5 and 10 mg/kg (Q and SLN-Q); Ad: i.p.; Duration: single	Light and dark chamber test; Partition preference test; Three horizontal compartment test; Spectroscopic method (GSH, TBARS, AChE levels)	Solid lipid nanoparticle of quercetin (SLN-Q) attenuated neurocognitive impairments along with amelioration of oxidative biomarker changes
Lu Y (2018) [15]	APP/PS1 transgenic AD mice (13 months old)	Dosage: 2 mg/g diet; Ad: p.o.; Duration: 9 or 13 months	Behavioral test (Morris Water Maze); Immunostaining (GFAP, 6E10); WB (APP, CTFβ, GFAP, etc); RT-qPCR (BACE1, PS1, Hevin, SPARC, Smad2, STAT3)	Ameliorated cognitive dysfunction only during early-middle stage of AD (astrogliosis↓, Aβ↓)
Karimipour M (2019) [16]	Aβ-injection rats AD model (Adult male Wistar rats, 350–400 g)	Dosage: 40 mg/kg/day; Ad: p.o.; Duration: 1 month	Morris water maze behavioral test; Immunohistochemistry (BrdU, DCX); Immunostaining (BrdU/NeuN double positive cells); RT-qPCR (BDN, NGF, CREB and ERG-1)	Increased proliferating neural stem/progenitor cells, enhanced adult neurogenesis (BDNF, NGF, CREB and EGR-1 genes expression↑)
Paula PC (2019) [17]	Homozygous 3xTg-AD mice (male&female, 6 months old)	Dosage: 100 mg/kg/48 h; Ad: p.o.; Duration: 12 months	Behavioral test (Elevated plus maze, Morris Water Maze); Immunohistochemistry (Aβ, AT-8)	Reduced β-amyloidosis, decreased tauopathy in hippocampus and amygdala, affected cognitive recovery
Li Y (2019) [18]	Aβ-injection rats AD model (male Sprague– Dawley rats, 220–280 g)	Dosage: 100 mg/kg/day; Ad: p.o.; Duration: 18 days	Morris water maze behavioral test; Estimation of oxidative stress (MDA level, SOD, CAT and GSH activity); Immunohistochemistry for Aβ; WB (Nrf2 and HO-1)	Promoted reversal of neuronal damage, Improved cognitive memory (Aβ1-42↓, antioxidant activity and Nrf2/HO-1 pathway↑)

**Table 2 ijms-21-00493-t002:** Characteristics of different Alzheimer’s disease (AD) models.

Model	Mechanism	Main Uses of the Model	Disadvantage
Aβ-induced	Neurotoxicity of Aβ species	Studying Aβ peptide aggregation and deposition, and its acute toxic effect in AD	Not reproducing the progressive neurodegeneration process as an acute model
LPS-induced	Inducing proinflammatory mediators, activating astrocytes and microglia	Simulating neuroinflammation and synaptic/memory dysfunction of AD	Lack of Aβ plaque accumulation and NFT formation
PTZ-induced	Activating free radicals and apoptotosis, modulating neurotransmitters metabolisms	Simulating oxidative damage, motor impairment as well as memory dysfunction of AD	Not replicating the histological hallmarks of AD
Senescence acceleratedmouse	Naturally rapid aging mouse model	Studying the mechanism of age-related spatial learning and memory deficits	Short lifecycle not supporting long-term animal experiments
APP/PS1/tau- transgenic	Aβ accumulation, NFT formation in the brain	Studying the role of APP and tau protein in the development of AD	Lack of APP and tau metabolism changes

**Table 3 ijms-21-00493-t003:** Methodological quality of studies.

Study	A	B	C	D	E	F	G	H	I	J	Score
Patil CS (2003) [5]	NC	Y	NC	Y	NC	NC	NC	Y	Y	NC	4
Wang DM (2014) [6]	Y	Y	NC	NC	NC	Y	NC	Y	Y	Y	6
Hayakawa M (2015) [7]	NC	NC	NC	Y	NC	NC	NC	N	Y	Y	3
Sabogal- Guáqueta AM (2015) [8]	NC	Y	NC	Y	NC	NC	NC	N	Y	Y	4
Zhang X (2016) [9]	NC	Y	NC	Y	NC	NC	NC	N	Y	NC	3
Sun D (2016) [10]	NC	NC	NC	Y	NC	NC	NC	NC	Y	NC	2
Moreno LCGEI (2017) [11]	NC	Y	NC	Y	N	NC	NC	NC	Y	Y	4
Vargas-Restrepo F (2018) [12]	Y	Y	NC	Y	NC	Y	NC	NC	Y	Y	6
Khan A (2018) [13]	NC	N	NC	Y	N	NC	NC	Y	Y	Y	4
Rishitha N (2018) [14]	NC	NC	NC	Y	NC	Y	NC	Y	Y	NC	4
Lu Y (2018) [15]	NC	NC	NC	Y	NC	NC	NC	Y	Y	NC	3
Karimipur M (2019) [16]	Y	Y	NC	Y	NC	NC	NC	Y	Y	Y	6
Paula PC (2019) [17]	NC	Y	NC	Y	NC	NC	NC	N	Y	Y	4
Li Y (2019) [18]	NC	NC	NC	Y	NC	Y	NC	Y	Y	Y	5

Legend: A—random allocation sequence; B—similar baseline characteristics; C—allocation concealment; D—random housing; E—blinded intervention; F—random selection for outcome assessment; G—blinded assessment of outcome; H—incomplete outcome data; I—selective outcome reporting; J—other sources of bias. Y: yes; N: no; NC: unclear.

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
