# Peer review of "Quercetin in Animal Models of Alzheimer’s Disease: A Systematic Review of Preclinical Studies"

_ijms, 2020, doi:10.3390/ijms21020493_

Round 1
Reviewer 1 Report
This is manuscript is a review of the effects of Quercetin and its pharmacological mechanisms on different animal models of Alzheimer’s Disease (AD). The authors have performed a systematic literature review to retrieve articles investigating the neuroprotective effects of Quercetin in rodent models of AD. The authors describe a clear definition of inclusion and exclusion criteria to search the PubMed, Web of Science and Google Scholar platforms. A total of 14 articles were included and have been evaluated for their methodological quality. Part of the results found in the included articles were also used for meta-analyses in this current manuscript. The authors also provided a nice summary figure with the potential neuroprotective mechanisms of Quercetin effects on AD animal models based on these articles.
A few comments/suggestions to consider:
A general revision of the manuscript is needed to correct several typos and orthography errors. The figure legends could be more explanatory and include a brief summary of the approach used. Although, part of this description appears in the text, it would be helpful to also include it in the figure legends. Additionally, the figures have very small font size and their quality should be improved. In “2.2.2 Behavioral test analysis”, the authors included data from 4 (out of 9) studies that conducted MWM test with reported number of times the mice crossed the target quadrant, for their meta-analysis. Is there a reason why the data from the remaining 5 studies were not included in this analysis? Additionally, the authors mentioned that 5 articles in the discussion (line 205), instead of 4. Table 3’s legend should include the acronyms displayed in the. table. The authors should further discuss the results of their methodological quality assessment and how this could have impacted the original articles’ conclusions.Author Response
Dear professor,
You find enclosed the revised manuscript according to your comments. The primary changes were detailed as follows:
The figure legends were added to explain the approach used in meta-analysis.
The reduplicated paragraph in the introduction was deleted.In the forest plot of figure 4, the mean difference of Ab baseline value was replaced with the ratio in order to standardize outcome.
The results of methodological quality assessment were further discussed.
The image resolution ratio was improved.grammar and typos errors were corrected.
Thanks very much for your instruction.
Best regards,
Xiaowen ZHANG

Reviewer 2 Report
The review entitled Quercetin in animal models of Alzheimer's disease: A systematic review of preclinical studies by Zhang, Chen, and Lu review important aspects of the therapeutic use of quercetin in a variety of animal models of Alzheimer disease and its potential use in future clinical studies in humans.
The authors analyze data regarding animal behavior, Ab levels and Ab IHC in different brain regions. In the end, authors discuss findings and their relation to neurodegeneration, neuroinflammation, oxidative stress, and mitochondrial dysfunction observed in such animal models.
The manuscript is well written and has a straightforward logical narrative. Performed statistical analysis is correct and the discussion covers the potential benefits of human use in the clinic as well as shortcomings and caveats of research available in animal models.
In my opinion, the manuscript is well suited for publication. I only have minor comments the authors might want to address to improve manuscript overall quality:
On page 2, line 53: This paragraph is repetitive. My suggestion is to be excluded since the previous paragraph already pointed out the link on pharmacological mechanisms and the potential of quercetin in future preclinical tests.
On page 9, Forest plot: Since the baseline values for Ab measurements by IHC are so variable among studies, my suggestion is to use a ratio instead of mean difference in order to standardize outcomes and allow better comparison between groups.
On page 10, line 148: Figure 5 instead of 3.
Author Response
Dear professor,
You find enclosed the revised manuscript according to your comments. The primary changes were detailed as follows:
The figure legends were added to explain the approach used in meta-analysis.
The reduplicated paragraph in the introduction was deleted.In the forest plot of figure 4, the mean difference of Ab baseline value was replaced with the ratio in order to standardize outcome.
The results of methodological quality assessment were further discussed.
The image resolution ratio was improved.grammar and typos errors were corrected.
Thanks very much for your instruction.
Best regards,
Xiaowen ZHANG
